

# PM2.5/PM10 Ratio Prediction Based on a Long Short-term Memory Neural Network in Wuhan, China

**Xueling Wu** [*]**, Ying Wang, Siyuan He, Zhongfang Wu**

Institute of Geophysics and Geomatics, China University of Geosciences, Wuhan 430074, China

* Corresponding author. E-mail: snowforesting@163.com; Tel: +86-27-67883251; Fax: +86-27-67883251

**Abstract**

Air pollution is a serious and urgent problem in China, and it has a great impact on the lives of residents and urban development. The particulate matter (PM) value is usually used to indicate the degree of air pollution. In addition to PM2.5 and PM10, the use of the PM2.5/PM10 ratio as an indicator and assessor of air pollution has also become more widespread. This ratio reflects the air pollution conditions and pollution sources. In this paper, a better composite prediction system was proposed that aimed at improving the accuracy and spatio-temporal applicability of PM2.5/PM10. First, the aerosol optical depth (AOD) in 2017 in Wuhan was obtained based on Moderate Resolution Imaging Spectroradiometer images, with a 1 km spatial resolution, by using the Dense Dark Vegetation method. Second, the AOD was corrected by calculating the planetary boundary layer height and relative humidity. Third, the coefficient of determination of the optimal subset selection was used to select the factor with the highest correlation with PM2.5/PM10 from meteorological factors and gaseous pollutants. Then, PM2.5/PM10 predictions based on time, space, and random patterns were obtained by using 9 factors (the corrected AOD, meteorological data and gaseous pollutant data) with the long short-term memory (LSTM) neural network method, which is a dynamic model that remembers historical information and applies it to the current output. Finally, the LSTM model prediction results were compared and analysed with the results of other intelligent models. The results showed that the LSTM model had significant advantages in the average, maximum and minimum accuracies and the stability of PM2.5/PM10 prediction.

**Keywords:** Air pollution · PM2.5/PM10 · MODIS · AOD · LSTM

## 1. Introduction



Aerosols are a general term for solid and gas particles suspended in air. Aerosols can have an important impact on
regional and global atmospheric environments, climates, and ecosystems and have long been an important issue in global
environmental change research (Crutzen and Andreae, 1990). Particulate matter (PM) is usually separated and
categorized based on its aerodynamic diameter, and the most widely monitored particles are PM10 and PM2.5. PM10 is
primarily produced by natural processes, such as resuspending local soils, sandstorms, and roadside dust, and various
industrial processes. Particles with an aerodynamic particle size not exceeding 2.5 μm are called fine PM (PM2.5), which
mainly derive from anthropogenic emissions. Anthropogenic combustion products come from transportation and energy
production and are particularly important for environmental policy and public health research (Pope and Dockery, 2006;
Xie et al., 2011). Infectious disease research shows that there is a significant consistency between the PM2.5
environmental quality concentration and adverse effects on human health (Lelieveld et al., 2015). PM2.5 mainly causes
damage to the respiratory and cardiovascular systems, including coughing, difficulty breathing, lowered lung function,
and aggravated asthma, causing chronic bronchitis, arrhythmia, non-fatal heart disease, and premature death of patients
with cardiopulmonary disease (Wu et al., 2011; Jia et al., 2012). In addition, since the scattering extinction contribution
of PM2.5 particles accounts for 80% of the extinction of the atmosphere, the concentration of PM2.5 is a key factor in
determining the visibility of the atmosphere. In view of the importance of aerosols and near-surface atmospheric PM2.5
to regional and global climates and environments, quantitative and accurate observations using a variety of observation
methods have become a hot research topic domestically and internationally (Dominici et al., 2006). Since fine and coarse
particles come from different sources, the PM2.5-PM10 scale model has different physicochemical properties, which can
not only distinguish the type of aerosol in the PM but also provide the mixing ratio of dust and artificial aerosols
(Sugimoto et al., 2015). For the research conducted in an urban area of northwestern China, PM10 and PM2.5
concentration data were collected to reveal the spatial-temporal behaviour of local PM and mineral dust fractions
(Qingyu et al., 2018).



The aerosol optical depth (AOD) is defined as the integral of the extinction coefficient of a medium in the vertical
direction, which describes the effect of aerosols on light reduction. A study conducted by Hidy in 2009 indicated that the
estimation of the PM2.5 concentration near the ground by satellite remote sensing AOD has great research potential
(Hidy, 2009). The advantage is that satellite remote sensing data are generally standardized data with high reliability and
a wide spatial coverage, providing wide-area, spatially continuous and real-time monitoring information for regional and
global PM2.5 air quality assessment. There are many ways to obtain the AOD from remote sensing data.
AOD products can be produced by many satellite sensors, such as the Geostationary Operational Environmental
Satellite (GOES) (Prados et al., 2007), Advanced Very High Resolution Radiometer (AVHRR) (Gao et al., 2016), and
Moderate Resolution Imaging Spectroradiometer (MODIS) (Levy et al, 2013). MODIS data are one of the most widely
used data sources for deriving ground PM2.5 concentrations with AOD (Hu et al., 2014). There are many ways to obtain
AOD through MODIS data. For example, Yang et al. used the data collected by Landsat 8 satellite images to retrieve the
AOD in Beijing by means of the Dark Target method and the visible near-infrared atmospheric correction method. The
accuracy was verified by the Aerosol Robotic Network (AERONET) observation data (Ou et al., 2017). The Dark Blue
AOD retrieval method was used to complement the Dark Target results by retrieving the AOD over bright arid land
surfaces, such as deserts (Sayer et al., 2013). In addition, a new method that considers bidirectional reflectance of the
surface was proposed, which is suitable for calculating the AOD in arid or semi-arid regions (Xinpeng et al., 2018).
Although the relationship between the AOD and PM has been proven by many scholars, since the PM concentration
level is usually measured at the surface, the correlation between them is affected by the planetary boundary layer height
(PBLH) and relative humidity (RH). When studying the seasonal PM10-AOD correlation in northern Italy, Arvani et al.
found that the introduction of PBLH and RH correction can significantly improve the bin-averaged PM AOD correlation
(Arvani et al., 2016). After the vertical and RH correction methods were applied to the air quality station in Beijing, the
determination coefficient $R^2$ of the AOD and PM10 increased by 0.13, and the correlation between the AOD and PM2.5





increased from 0.48 to 0.62 (Wang et al., 2010). These calibration methods usually require the use of meteorological data
to perform the calculation, and the addition of meteorological data to the evaluation of PM concentration can give better
results. For instance, Jung et al. joined meteorological data to obtain an improved model of the surface PM2.5 from 2005
to 2015 to estimate the PM concentration for the entire main island of Taiwan (Jung et al., 2017).

Many statistical models have been used for the ground PM estimation of AOD and other predictors, such as linear

regression models, random forest models, neural network models, and generalized additive models. However, with the
introduction of new intelligent models, the traditional regression model reflects the inability to balance time, space and
random precision. One way to overcome these limitations is the long short-term memory (LSTM) model. The LSTM
network is ideal for learning from experience so that time series can be classified, processed, and predicted with very
long unknown time lags between important events. In the study of PM2.5 monitoring and prediction in smart cities,
Chiou-Jye et al. proposed that the prediction accuracy of the convolutional neural network (CNN)-LSTM model is the
highest compared to the prediction accuracies of several other classic machine learning methods (Chiou-Jye and
Ping-Huan, 2018). Xiang et al. used the LSTM model to automatically extract inherent useful features from historical air
pollutant data to obtain a more efficient multi-scale prediction framework (Li et al., 2017).

This paper used a total of 59 AOD results for all of 2017 by the Dense Dark Vegetation (DDV) method using

MODIS level-2 data of Wuhan with a spatial resolution of 1 km. Since there were only 10 air quality stations in Wuhan,
to ensure accuracy, the AOD values were extracted at the air quality station site, and the integration of the AOD, air
pollutants, and meteorological data was also based on the station site. AOD* was obtained by correcting AOD using the
PBLH and RH. Then, the $R^2$-based optimal subset selection method was used to select the most relevant factor for
PM2.5/PM10 from the meteorological factors and air pollutants. Finally, the space and time scales and random
PM2.5/PM10 predictions were determined and performed, respectively, via the LSTM model, and the prediction results
of the LSTM model and other classic models were compared and analysed.



## 2. Study area


Wuhan is the provincial capital of Hubei Province. The administrative extent is between 113.683°E-115.083°E and
29.967°N-31.367°N, and the total area is 8494.41 km² (Zhou and Chen, 2018). The largest distance is between the
eastern and western parts of Wuhan and is 134 km, and the maximum distance from north to south is 155 km. Wuhan is
the city with the largest population, is the largest provincial capital city, has the most complicated road traffic and has the
most developed economy in the central part of the country (Jiao et al., 2017). The Yangtze River flows through Wuhan,
and there are hundreds of lakes in Wuhan. The terrain of Wuhan is mainly plains, with low levels in the middle of the
region and low mountains, hills and ridges to the south and north. The climate type is a humid, north subtropical
monsoon climate with high temperatures in summer, low temperatures in winter, and an annual average temperature of
15.9 °C. Sunshine hours and total radiation are also at high levels, and the annual average precipitation is approximately
1300 mm. June and August receive the most precipitation in Wuhan, and summer precipitation accounts for
approximately 40% of the annual rainfall. In recent years, the air quality in Wuhan has been improved. In 2017, the
number of days in which the annual air quality level was acceptable was 255 days, and the acceptability rate was 69.9%.
At the same time, the number of days with light pollution, moderate pollution, heavy pollution, and severe pollution was
86 days, 17 days, 6 days, and 1 day, respectively.

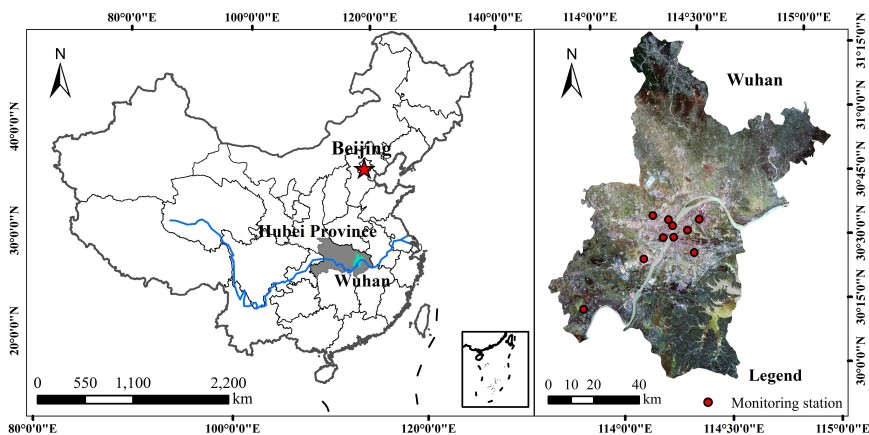






(A)                                                                          (B)

**Fig. 1** Location of the study area in China (A: map of China, B: map of Wuhan).
## 3. Methods
The data that our environmental monitoring station can monitor are only real-time data. If we want to predict the
state of the air afterwards, we can use other relevant factors for reference. The AOD is an important parameter in the
study of atmospheric aerosols, which have a great relationship with PM. Gaseous pollutants are also a key factor in air
quality. In addition, changes in meteorological conditions have an impact on PM. Therefore, we used the air quality data
from the ground monitoring station as the inspection standard and extracted the values of these correlation factors with
the data from the monitoring site for verification. After retrieving the AOD with the MODIS images, the AOD values at
the monitoring site were extracted, and the values of the meteorological data were also interpolated at the same point.
Then, the AOD was corrected to obtain the AOD*, and gaseous pollutant data at the monitoring site were added. The
best set that predicted air quality was selected, and machine learning techniques were used to obtain models that can
make space and time series predictions (Fig. 2).

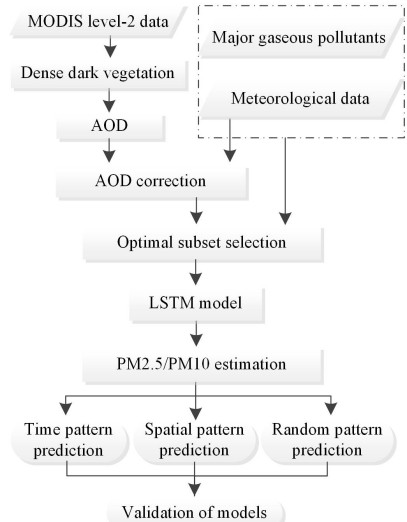






**Fig. 2** A flow chart of the research process.
3.1 AOD retrieval
MODIS is an important sensor on the Terra and Aqua satellites. The Terra satellite is a morning star, passing from
north to south at approximately 10:30, and Aqua acts as an afternoon star, moving from south to north at 13:30. Wuhan is
located in the central and eastern parts of Hubei Province at the southeast corner of the h27v05 frame; therefore, we
chose to use the images collected by Terra because of its higher image quality. The MODIS data have 36 spectral bands,
ranging from 0.4 μm to 14.4 μm, of which 7 bands can be used to retrieve the AOD, while the best bands for over-land
aerosol retrieval are 0.47 μm, 0.66 μm, and 2.12 μm, especially in areas with dense vegetation. We downloaded the
MOD02_L1B data for the region in Wuhan in 2017 via the website (https://ladsweb.modaps.eosdis.nasa.gov) and
removed a number of days with a large amount of clouds, finally obtaining 59 images with a spatial resolution of 1 km.
According to the DDV method (Li et al., 2014), after radiation correction, geometric correction, angle data resampling,
and angle data geometric correction and synthesis, cloud detection processing was performed; then, a lookup table file
was generated according to the "6S" atmospheric radiation model, and the AOD was acquired (Fig. 3). After verifying
with the MOD04_L2 aerosol product data released by the National Aeronautics and Space Administration (NASA), the
results of the retrieval were considered valid and used later. Fig. 4 shows the results of the AOD retrieval on July 18[th].

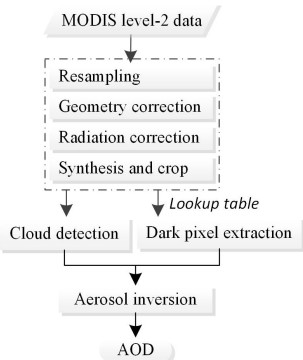


**Fig. 3** A flow chart of the AOD retrieval.





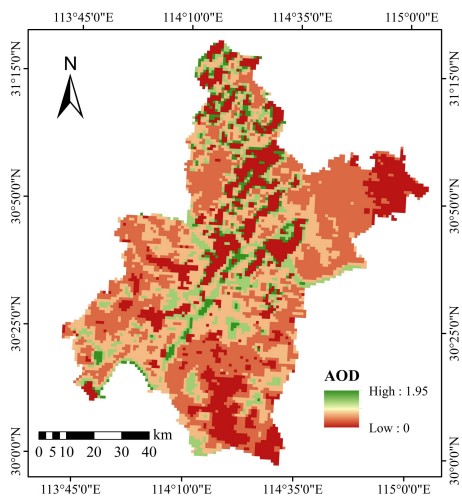


**Fig. 4** AOD retrieval on July 18th.


3.2 Ground-level air quality and gaseous pollutant data

The Ministry of Ecology and Environment of China has established 10 national environmental quality control

stations in Wuhan. The shortest distance between points exceeds 3 km, and the average distance exceeds 10 km. Each
station continuously collects hourly average concentration values of PM2.5, PM10, $SO_2$, $NO_2$, $O_3$, and CO and publishes
the daily average concentration values. The calculations in this paper were based on these daily averaged data. The
monthly average concentration data of PM2.5, PM10, and gaseous pollutants obtained from these data in 2017 are shown
in Table 1. During the year, the trends in PM2.5 and PM10 were roughly the same. From February to April, the values
dropped rapidly. From April to May, both experienced a small increase, and there were decreases from May to July. The
concentration of PM2.5 continued to rise after July, and the growth rate became larger. The concentration of PM10 also
increased after July but decreased between September and October. $NO_2$ is mainly derived from the high-temperature
combustion process of fossil fuels. The combustion of nitrogen-containing fuels (such as coal) and nitrogen-containing
chemicals can directly release $NO_2$. In general, motor vehicle emissions are one of the main sources of urban $NO_2$. $SO_2$ is
a ubiquitous pollutant in cities. The $SO_2$ in the air mainly comes from the industrial production of thermal power



generation and other industries, such as the combustion of fixed-source fuels; the production of non-ferrous metals; the
production of steel, chemical, and sulfur plants; and discharge from small heating boilers and civil coal furnaces. Natural
processes, such as volcanic activity, also emit a certain amount of $SO_2$. CO is a colourless, odourless, flammable, and
toxic gas that is a product of the incomplete combustion of carbonaceous fuels. The concentrations of $SO_2$, $NO_2$, and CO
showed regularity. The concentration in summer was the lowest, followed by spring and autumn, and the highest was in
winter. The lowest value was in June or July, and the highest was in December. $O_3$ is a representative pollutant for
photochemical smog, which is formed and enriched by nitrogen oxides and hydrocarbons in the air under intense sunlight
and through a series of complex atmospheric chemical reactions. Although $O_3$ in the upper stratosphere has important
anti-radiation protection for life on Earth's, $O_3$ at low altitudes in cities is a very harmful pollutant. The trend in the $O_3$
concentration was different, where the winter value was low and then increased in spring with time. In summer, the $O_3$
concentration fluctuated at a higher level and decreased in autumn.
**Table 1** Monthly average concentrations of PM2.5, PM10, and gaseous pollutants in 2017.

| Month | PM2.5 ($\mu g/m^3$) | PM10 ($\mu g/m^3$) | $SO_2$ ($\mu g/m^3$) | $NO_2$ ($\mu g/m^3$) | $O_3$ ($\mu g/m^3$) | CO ($mg/m^3$) |
|---|---|---|---|---|---|---|
| Jan | 99.48 | 147.26 | 26.66 | 48.20 | 36.86 | 1.40 |
| Feb | 121.17 | 167.42 | 16.63 | 46.01 | 36.13 | 1.44 |
| Mar | 59.44 | 145.11 | 27.04 | 51.88 | 60.96 | 1.11 |
| Apr | 41.27 | 93.87 | 16.07 | 38.35 | 93.18 | 0.93 |
| May | 52.85 | 107.95 | 12.00 | 40.15 | 125.30 | 0.93 |
| Jun | 27.80 | 55.35 | 4.82 | 25.45 | 102.20 | 0.81 |
| Jul | 24.23 | 53.13 | 6.05 | 17.77 | 107.92 | 0.62 |
| Aug | 27.37 | 65.09 | 11.07 | 24.47 | 73.24 | 1.04 |
| Sep | 36.20 | 87.85 | 19.11 | 40.55 | 139.25 | 1.33 |
| Oct | 39.07 | 77.20 | 13.65 | 43.64 | 54.00 | 1.10 |
| Nov | 90.88 | 134.91 | 21.53 | 62.36 | 54.28 | 1.19 |
| Dec | 111.15 | 148.29 | 27.06 | 70.21 | 21.78 | 1.50 |

3.3 Meteorological data

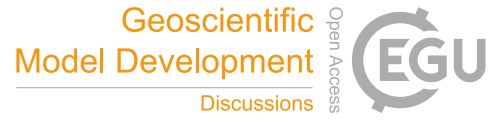

The quality of air is closely related to meteorological conditions. The meteorological data obtained in this paper
derive from the National Meteorological Information Center of China's National Meteorological Information Network
(http://data.cma.cn/site/index.html) and includes average rainfall, evaporation capacity, RH, sunshine intensity, average
surface temperature, average wind velocity, average air pressure, and average temperature. The data obtained were daily
average data in 2017. A total of 5 meteorological stations exist near the Wuhan area. To obtain meteorological data near
the air quality monitoring stations, data from the meteorological stations needed to be interpolated. After comparing the
kriging, natural neighbour, spline, and inverse distance weighted methods, we found that the results acquired by setting
12 interpolation points and using the spherical model of the kriging method were more suitable for the study area. The
kriging method is a multi-step process that includes the exploratory statistical analysis of the data, the modelling of
variograms, the creation of surfaces, and the study of varying surfaces. The monthly averages of the meteorological data
at all of the calculated sites are shown in Table 2. The seasonal changes reflected by several meteorological data results
were more obvious. The average surface temperature and average temperature showed a higher trend in summer and a
lower trend in winter. The average air pressure had a completely opposite trend. The sunshine intensity and evaporation
capacity were lower in winter and fluctuated in the other three quarters. The rainfall was concentrated in summer and
autumn, while the average wind velocity and RH had no obvious seasonal characteristics.
**Table 2** Monthly averages of the meteorological data.

| Month | Average rainfall (0.1 mm) | Evaporation capacity (0.1 mm) | Average surface temperature (0.1℃) | Average air pressure (0.1 hPa) | Relative humidity (-1%) | Sunshine intensity (0.1 h) | Average temperature (0.1℃) | Average wind velocity (0.1 m/s) |
|---|---|---|---|---|---|---|---|---|
| Jan | 0.00 | 18.09 | 62.19 | 10230.27 | 63.91 | 58.06 | 47.78 | 16.51 |
| Feb | 38.84 | 19.55 | 108.27 | 10151.31 | 72.03 | 24.23 | 103.45 | 29.35 |
| Mar | 0.00 | 29.34 | 140.11 | 10166.74 | 64.14 | 94.10 | 115.67 | 14.52 |
| Apr | 0.00 | 35.81 | 211.98 | 10103.29 | 69.60 | 105.93 | 181.67 | 16.16 |
| May | 0.00 | 36.81 | 288.18 | 10062.96 | 66.83 | 103.69 | 240.91 | 10.72 |
| Jun | 30.49 | 37.48 | 289.44 | 10002.23 | 84.54 | 64.80 | 261.32 | 18.69 |



| | | | | | | | |
|------|-------|-------|--------|----------|-------|--------|--------|-------|
| Jul | 2.33 | 57.25 | 366.30 | 10011.06 | 70.70 | 112.87 | 317.36 | 22.14 |
| Aug | 24.15 | 37.88 | 318.01 | 10017.01 | 81.09 | 84.67 | 296.38 | 18.88 |
| Sep | 0.00 | 45.47 | 289.04 | 10093.00 | 69.64 | 106.04 | 242.16 | 19.61 |
| Oct | 20.54 | 19.50 | 199.33 | 10138.21 | 84.03 | 61.31 | 176.99 | 11.60 |
| Nov | 0.00 | 21.36 | 157.65 | 10180.33 | 75.21 | 85.71 | 131.89 | 13.28 |
| Dec | 0.00 | 15.80 | 59.94 | 10222.16 | 67.78 | 76.57 | 42.91 | 9.12 |

## 4. Methods

### 4.1 AOD correction

The PBLH refers to the thickness of the planetary boundary layer and is an important physical parameter for

numerical atmospheric models and environmental evaluations (Su et al., 2018). The PBLH is calculated by a commonly

used national standard method in China. The national standard method is performed according to the method specified in

the Chinese national standard GB/T13201-91. This method assumes that the thermal conditions of the near-surface layer

depend, to a large extent, on the degree of ground heating and cooling. This method takes into account the thermal and

dynamic factors and quantifies the solar elevation angle, cloud volume, and wind speed. Then, according to the specified

local parameters, the atmospheric stability is classified into A, B, C and D categories according to the Pasquill stability

classification:

$$h = \frac{a_s U_{10}}{f} \tag{1}$$

When the atmospheric stability is E and F:

$$h = \frac{b_s \sqrt{U_{10}}}{f} \tag{2}$$

$$f = 2\Omega \sin \varphi \tag{3}$$

where $h$ (m) is the thickness of the mixing layer; $U_{10}$ (m*s$^{-1}$) is the average wind velocity at a height of 10 m, which

is 6 m*s$^{-1}$; $a_s$ and $b_s$ are the mixing layer coefficients; $f$ is the ground rotation parameter; $\Omega$ is the ground rotation angular

velocity, with a value of $7.29 \times 10^{-5}$ rad*s$^{-1}$; and $\varphi$ (°) is the geographic latitude.





The aerosol hygroscopic growth factor f(RH), where RH is the relative humidity, describes the extent to which the
aerosol extinction cross section or scattering coefficient increases with increasing RH, depending on a variety of factors,
such as the temperature absorption properties of the aerosol (Cai et al., 2018). The common formula for calculating f(RH)
is:

$$f(RH) = 1 / (1 - RH / 100) \qquad (4)$$

Since the parameters describing atmospheric physical conditions, such as air pressure, atmospheric temperature and
atmospheric humidity change, exist much more in the vertical than horizontal direction, it is often assumed that the
atmosphere has a structure in which the horizontal direction is uniform, and the vertical direction is layered. For the
single homogeneous distribution of spherical aerosol particles, the near-surface particle concentration can be obtained by
measuring a dry air sample. The expression is as follows:

$$PM = \frac{4}{3} \pi \rho \int r^3 n(r) dr \qquad (5)$$

where $\rho$ (g/m$^3$) is the average density of the particles and $n(r)$ is the particle spectral distribution function under
ambient humidity, which is related to the particle size.
Given the wavelength of the radiation, the aerosol optical thickness from the ground to a height of H can be
expressed as:

$$AOD = \pi \int_0^H \int_0^\infty Q_{ext,amb}(r) n_{amb}(r) r^2 dr dz \qquad (6)$$

To convert $Q_{ext,amb}$ under ambient humidity to $Q_{ext,dry}$ under dry conditions, a hygroscopic growth factor f(RH) is
required. This factor represents the ratio of normalized particle scattering efficiency under ambient RH and dry
conditions and is a function of humidity:





$$AOD = \pi f(RH) \int_0^H \int_0^\infty Q_{ext,dry}(r)n(r)r^2 dr dz \tag{7}$$

A normalized particle scattering efficiency $Q_{ext}$ and a parameterized expression of the effective radius $r_{eff}$ are
introduced for replacement in the above formula:

$$Q_{ext} = \frac{\int r^2 Q_{ext}(r)n(r)dr}{\int r^2 n(r)dr} \tag{8}$$

$$r_{eff} = \frac{\int r^3(r)n(r)dr}{\int r^2 n(r)dr} \tag{9}$$

Finally, the relationship between the AOD and near-surface PM2.5 mass concentration is introduced:

$$AOD = PM2.5Hf(RH)\frac{3Q_{ext,dry}}{4\rho r_{eff}} = PM2.5HS \tag{10}$$

where $S$ ($m^2 g^{-1}$) represents the specific extinction efficiency of the aerosol under ambient humidity conditions. $H$
stands for aerosol elevation. In practice, the PBLH approximation is often used instead of $H$. According to the above
relationship between the AOD and PM2.5, it can be inferred that if the AOD is corrected by the factors PBLH and f(RH),
the corrected AOD*, that is, AOD/(PBLH*f(RH)), is expected to have better correlation with PM. Taking the monthly
average value as an example, the parameters PBLH and f(RH) used by the AOD correction algorithm and the corrected
AOD* are shown in Table 3. The monthly average data of PM2.5/PM10, AOD and AOD* are shown in Fig. 5. In fact,
after calculating the linear correlations of the AOD and AOD* with PM2.5/PM10, the correlation increased from 0.838
to 0.873.
**Table 3** Monthly average AOD, PBLH, f(RH), and AOD*.

| Month | AOD ($\times 10^{-1}$) | PBLH | f(RH) | AOD* ($\times 10^{-4}$) |
|---|---|---|---|---|
| Jan | 12.610 | 428 | 4.00 | 7.366 |
| Feb | 12.343 | 444 | 3.85 | 7.221 |
| Mar | 9.200 | 461 | 4.00 | 4.989 |



| | | | | |
|------|--------|-----|------|--------|
| Apr | 5.192 | 713 | 4.00 | 1.820 |
| May | 5.625 | 686 | 4.00 | 2.050 |
| Jun | 4.000 | 631 | 5.00 | 1.268 |
| Jul | 3.895 | 686 | 5.56 | 1.021 |
| Aug | 5.083 | 686 | 5.26 | 1.409 |
| Sep | 6.375 | 741 | 4.35 | 1.978 |
| Oct | 4.964 | 395 | 4.00 | 3.142 |
| Nov | 10.06 | 412 | 3.85 | 6.345 |
| Dec | 15.263 | 412 | 3.57 | 10.377 |

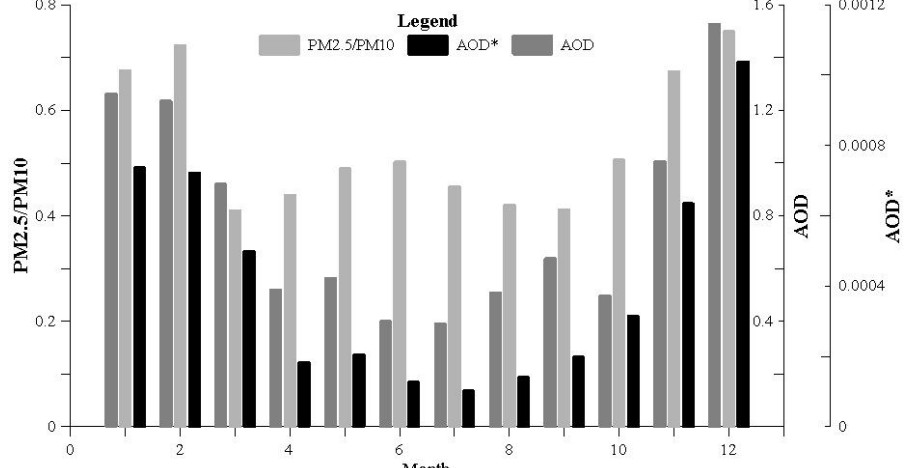


**Fig. 5** A bar chart of monthly average PM2.5/PM10, AOD and AOD*.

## 4.2 Selection factors

When choosing a subset, the choice of independent variables should be practical. How to choose the best subset of

variables to establish a better regression equation has been a hot research topic. An optimal way to choose a regression

equation is to combine all of the independent variables with the dependent variable to establish all possible equations and

then select one of the best-performing subsets from all possible equations. This is called the optimal subset method. The

optimal subset method can determine an optimal regression equation from all possible subsets via some criteria and has

been widely used in weather and climate predictions. Using the correlation coefficient $R^2$ as the evaluation index and the



optimal subset of PM2.5/PM10 as the dependent variable, the highest $R^2$ is 0.461. The independent variables in the
subset are AOD*; average rainfall; evaporation capacity; RH; sunshine intensity; average wind velocity; and $SO_2$, CO,
and $O_3$ concentrations. The factors selected by the optimal subset method are shown in Table 4. The symbol "√" indicates
that the factor is selected.
**Table 4** Factors selected by the optimal subset method.

| Factors \ $R^2$ | 0.461 | 0.460 | 0.460 | 0.457 | 0.455 | 0.455 | 0.454 | 0.453 | 0.452 | 0.452 |
|---|---|---|---|---|---|---|---|---|---|---|
| CO | √ | √ | √ | √ | √ | √ | √ | √ | √ | √ |
| Average rainfall | √ | √ | √ | √ | √ | | √ | √ | √ | √ |
| Evaporation capacity | √ | √ | √ | √ | √ | √ | √ | √ | √ | √ |
| Relative humidity | √ | √ | √ | | √ | | √ | √ | √ | √ |
| Sunshine intensity | √ | √ | √ | √ | √ | √ | √ | √ | | √ |
| Average wind velocity | √ | √ | | √ | √ | √ | √ | √ | √ | √ |
| AOD* | √ | √ | √ | √ | √ | √ | √ | √ | √ | √ |
| SO₂ | √ | √ | √ | √ | √ | √ | √ | √ | √ | √ |
| O₃ | √ | √ | √ | √ | √ | √ | √ | √ | √ | √ |
| Average air pressure | | √ | | √ | √ | √ | | | | |
| Average surface temperature | | | | √ | √ | √ | | | | √ |
| Average temperature | | | | | | | √ | | | |
| NO₂ | | | | | | | | √ | | |

4.3 RNNs and the LSTM model
The recurrent neural network (RNN) is a powerful deep neural network that uses its internal memory to process
input sequences with any timing. In the RNN model, compared with the common multi-layer neural network, the
interconnection layer is added between the nodes of the hidden layer, and the directional loop is formed by the
connection between the hidden layer neural units; then, the internal state of the network is created, and the dynamic time



series behaviour is presented (Bao and Zeng, 2013). The RNN can handle any sequence length in principle, but in an
actual situation, the standard RNN model cannot store sequence information about the past and lacks the ability to
establish remote structure connections. This kind of "forgetting" limitation cannot record long-term information. Thus,
these networks are more prone to instability when generating sequences, resulting in a time dependency problem. This
problem is not unique to RNNs but exists in almost all generation models. The LSTM model is a network that is used to
address long-term time-dependent dependencies. It is a time-RNN suitable for processing and predicting important
events with relatively long intervals and delays in time series (Weninger et al., 2014).

The key to distinguishing the LSTM model from the traditional RNN is that the traditional RNN has only one

hidden layer output value state $h$, and $h$ changes with the convolution process and is insensitive to long-term or
long-distance events. The LSTM model adds a unit state $c$ to store the long-term status. The calculation process after
adding $c$ is shown in Fig. 6:


**Fig. 6** The calculation process of unit $c$ in the LSTM model.

where $x$, $h$, and $c$ are vectors. At time $t$, there are three inputs to the LSTM: the input value $x_t$ of the current time

network, the output value $h_{t-1}$ of the LSTM model at the previous time, and the unit state $c_{t-1}$ of the previous time. The
two outputs of the LSTM are the current time LSTM output value $h_t$ and the current state of the unit $c_t$.

The key point of the LSTM model is how to control the state $c$. The idea of the LSTM model is to use three control

switches to control it. The first switch control continues to store $c$, the second switch control inputs the current state to $c$,
and the third switch controls whether $c$ is the current output of the LSTM model. The switches implemented in the



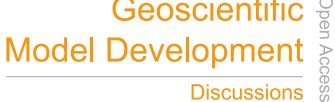

algorithm are known as "gates", which are fully connected layers whose input is a vector, and the output is a real vector
between 0 and 1 (Srivastava and Lessmann, 2018). Assuming $W$ is the weight vector of a gate and $b$ is the bias value,
then the gate can be expressed as:

$$g(x) = s(Wx + b) \tag{11}$$

These three gates are defined as follows:

$$i_t = \sigma(W_i * [h_{t-1}, x_t] + b_i) \tag{12}$$

$$f_t = \sigma(W_f * [h_{t-1}, x_t] + b_f) \tag{13}$$

$$o_t = \sigma(W_o * [h_{t-1}, x_t] + b_o) \tag{14}$$

where $i_t$, $f_t$, and $o_t$ are the values of the input, forget, and output gates, respectively; $\sigma$ is the activation function; and

$b_i$, $b_f$, and $b_o$ are their respective bias values. The structure of the LSTM model is shown in Fig. 7. The inputs are in terms
of time, space and randomness, and the outputs are their results.

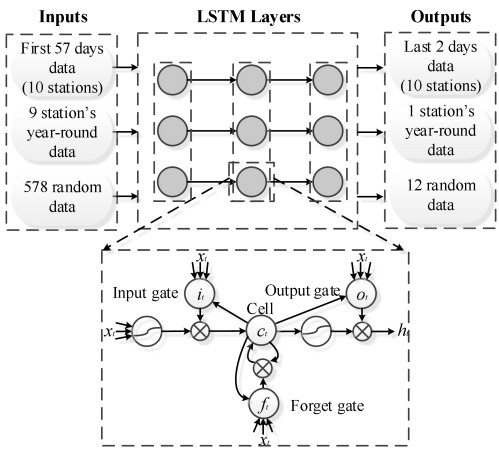


**Fig. 7** Architecture of the LSTM model.
**5.    Results and discussion**



To determine the appropriate number of layers for the LSTM method, we divided the training data set into two parts:
80% of the data were used as the training sample for modelling, and 20% of the data were used as the verification sample.
We tried to use various layers for the comparison. After obtaining the results of various layers, we found that the results
obtained using the four-layer LSTM structure were the best, with the first three layers being the LSTM layer and the last
layer being the dense layer. Because the LSTM uses the activation function as the gate, the outputs of the gates must be
between 0 and 1, and the output ranges of both types of activation functions must be satisfied. We determined that the
activation function for setting the forget gate and the input gate was defined as a sigmoid function. The best activation
function for outputting the results was the tanh function.
5.1 Time pattern prediction
Using the input of the first 57 days in the 2017 data from 10 sites, there were 570 input samples, and the data used
to verify the model were from the last two days in 2017. These two days were December 25 and December 31. In winter,
with a high PM2.5/PM10 value, the ratios were more concentrated above 0.6. We compared the prediction results of the
LSTM model with the multi-layer perceptron (MLP), back propagation (BP) artificial neural network, support vector
machine (SVM), and chi-squared automatic interaction detector (CHAID) decision tree models. Then, we calculated the
error rate between the predicted value and the measured value (Table 5). Among the five algorithms, the average error of
the LSTM model was the smallest, 15.1704, and its minimum error was also the smallest, only 0.877, but its maximum
error value was larger than the BP and SVM maximum errors values. The MLP method had the worst predictions,
whether in terms of the average error, maximum error or minimum error. It seemed that the MLP method was not
suitable for predictions in terms of air quality time series. The BP network method and the SVM had similar prediction
results; the average error was not too large, and the maximum error value was smaller than that of the LSTM, while the
minimum error value was larger. Although the average error of the CHAID model was small, the minimum error and the
maximum error values were both bad. None of the five prediction methods could accurately predict the case where the





PM2.5/PM10 value was greater than 0.9. The maximum value that the LSTM was able to predict was 0.8848. The
predicted maximum values of the MLP, BP, SVM, and CHAID were 0.7606, 0.8321, 0.8568, and 0.8206, respectively.
**Table 5** The results and relative error rates of the time pattern predictions.

| Measured value | Predicted value | | | | | Relative error rate (%) | | | | |
|---|---|---|---|---|---|---|---|---|---|---|
| | LSTM | MLP | BP | SVM | CHAID | LSTM | MLP | BP | SVM | CHAID |
| 0.8212 | 0.7682 | 0.7329 | 0.7786 | 0.6698 | 0.4853 | 6.4540 | 10.7526 | 5.1875 | 18.4364 | 40.9036 |
| 0.7436 | 0.6910 | 0.6526 | 0.6961 | 0.7841 | 0.4853 | 7.0737 | 12.2378 | 6.3878 | 5.4465 | 34.7364 |
| 0.6629 | 0.5962 | 0.4624 | 0.7074 | 0.8353 | 0.6753 | 10.0618 | 30.2459 | 6.7129 | 26.0069 | 1.8706 |
| 0.6950 | 0.6297 | 0.5955 | 0.6850 | 0.5628 | 0.6753 | 9.3957 | 14.3165 | 1.4388 | 19.0216 | 2.8345 |
| 0.7816 | 0.6102 | 0.5134 | 0.6871 | 0.8092 | 0.5145 | 21.9294 | 34.3142 | 12.0906 | 3.5312 | 34.1735 |
| 0.6311 | 0.6795 | 0.6608 | 0.5864 | 0.7032 | 0.6487 | 7.6691 | 4.7061 | 7.0829 | 11.4245 | 2.7888 |
| 0.7959 | 0.4918 | 0.5211 | 0.6870 | 0.8568 | 0.6973 | 38.2083 | 34.5270 | 13.6826 | 7.6517 | 12.3885 |
| 0.8743 | 0.8487 | 0.7104 | 0.6474 | 0.7451 | 0.6973 | 2.9281 | 18.7464 | 25.9522 | 14.7775 | 20.2448 |
| 0.7204 | 0.4774 | 0.6087 | 0.8106 | 0.7446 | 0.8206 | 33.7313 | 15.5053 | 12.5208 | 3.3592 | 13.9089 |
| 0.9854 | 0.6031 | 0.7445 | 0.7154 | 0.6760 | 0.8206 | 38.7964 | 24.4469 | 27.4000 | 31.3984 | 16.7242 |
| 0.7079 | 0.7842 | 0.7606 | 0.8321 | 0.6089 | 0.7959 | 10.7784 | 7.4446 | 17.5449 | 13.9850 | 12.4311 |
| 0.9455 | 0.7127 | 0.7531 | 0.7064 | 0.7285 | 0.7959 | 24.6219 | 20.3490 | 25.2882 | 22.9508 | 15.8223 |
| 0.7200 | 0.4969 | 0.4701 | 0.6692 | 0.8172 | 0.6931 | 30.9861 | 34.7083 | 7.0556 | 13.5000 | 3.7361 |
| 0.8600 | 0.8848 | 0.5717 | 0.6192 | 0.6907 | 0.6931 | 2.8837 | 33.5233 | 28.0000 | 19.6860 | 19.4070 |
| 0.6571 | 0.6311 | 0.6055 | 0.7011 | 0.8522 | 0.5812 | 3.9568 | 7.8527 | 6.6961 | 29.6911 | 11.5508 |
| 0.9189 | 0.6849 | 0.6583 | 0.6195 | 0.7146 | 0.5812 | 25.4652 | 28.3600 | 32.5824 | 22.2331 | 36.7505 |
| 0.7640 | 0.7573 | 0.5281 | 0.6549 | 0.5406 | 0.7870 | 0.8770 | 30.8770 | 14.2801 | 29.2408 | 3.0105 |
| 0.9273 | 0.7777 | 0.5247 | 0.6354 | 0.7155 | 0.7870 | 16.1329 | 43.4164 | 31.4785 | 22.8405 | 15.1299 |
| 0.6277 | 0.6417 | 0.7458 | 0.7308 | 0.5392 | 0.6951 | 2.2304 | 18.8147 | 16.4250 | 14.0991 | 10.7376 |
| 0.8896 | 0.8075 | 0.6556 | 0.6685 | 0.6694 | 0.7534 | 9.2289 | 26.3040 | 24.8539 | 24.7527 | 15.3103 |
| Mean: | | | | | | 15.1704 | 22.5724 | 16.1330 | 17.7017 | 16.2230 |
| Maximum: | | | | | | 38.7964 | 43.4163 | 32.5824 | 31.3984 | 40.9036 |
| Minimum: | | | | | | 0.8770 | 4.7061 | 1.4388 | 3.3592 | 1.8706 |

5.2 Spatial pattern prediction





296 One station was used as the output to be predicted; the other nine sites were inputs, and the prediction results of the

297 spatial pattern were obtained. The output site is located in the southwest corner of Wuhan, which is the farthest from the

298 other stations, and the distance from the nearest station is 34.7 km. The relative error rates of the predicted results of the

299 five models are shown in Table 6. The average error rate of the LSTM model was still the lowest, along with the

300 maximum error value, which was much smaller than that of the other models. The minimum error rate of the LSTM

301 model was 0.1545%, which was not the lowest but was much smaller than the results of the SVM and CHAID model. In

302 addition, we also conducted experiments using one station located in the central area of Wuhan as the output. The results

303 of the LSTM model showed that the prediction results at this point were much better than those at the southwest point,

304 and the average error rate was 25.1664%.

305 **Table 6** The results and relative error rates of the spatial pattern prediction.

| Models | LSTM | MLP | ANN | SVM | CHAID |
|---|---|---|---|---|---|
| Mean: | 32.1585 | 37.6755 | 34.1333 | 34.0207 | 33.7718 |
| Maximum: | 160.3270 | 216.3275 | 222.9295 | 204.7317 | 230.1367 |
| Minimum: | 0.1545 | 0.1451 | 0.1124 | 0.9026 | 0.2396 |

309 ## 5.3 Random pattern prediction

310 The random pattern prediction randomly selected 12 data points as the outputs among all 590 data points. The

311 randomly selected measured data ranged from 0.2222 to 0.9843, covering the entire range of monitored values. After

312 calculating the prediction results and relative error rates of the five models, the average, maximum and minimum error

313 rates of the LSTM model were the smallest, and the results were significantly better than those of the other methods

314 (Table 7). The predictions for the maximum and minimum values were also relatively good. However, it could be found

315 that the prediction results obtained by these models were concentrated between 0.35 and 0.75, and the prediction results

316 of the minimum and maximum values were generally poor.



**Table 7** The results and relative error rates of the random pattern prediction.

| Measured value | Predicted value | | | | | Relative error rate (%) | | | | |
|---|---|---|---|---|---|---|---|---|---|---|
| | LSTM | MLP | BP | SVM | CHAID | LSTM | MLP | BP | SVM | CHAID |
| 0.5870 | 0.5723 | 0.5443 | 0.5762 | 0.6091 | 0.4928 | 2.5043 | 7.2743 | 1.8399 | 3.7649 | 16.0477 |
| 0.6213 | 0.7449 | 0.6402 | 0.6561 | 0.6826 | 0.6795 | 19.8938 | 3.0420 | 5.6012 | 9.8664 | 9.3675 |
| 0.9843 | 0.6650 | 0.4874 | 0.6247 | 0.6185 | 0.7422 | 32.4393 | 50.4826 | 36.5336 | 37.1635 | 24.5962 |
| 0.8000 | 0.6238 | 0.4500 | 0.4772 | 0.5231 | 0.4928 | 22.0250 | 43.7500 | 40.3500 | 34.6125 | 38.4000 |
| 0.4638 | 0.4656 | 0.4773 | 0.4773 | 0.5136 | 0.4928 | 0.3881 | 2.9107 | 2.9107 | 10.7374 | 6.2527 |
| 0.7010 | 0.6913 | 0.5697 | 0.6811 | 0.6675 | 0.6795 | 1.3837 | 18.7304 | 2.8388 | 4.7789 | 3.0670 |
| 0.2222 | 0.3502 | 0.5598 | 0.4292 | 0.3971 | 0.3737 | 57.6058 | 151.9352 | 93.1593 | 78.7129 | 68.1818 |
| 0.5929 | 0.7606 | 0.6807 | 0.6543 | 0.6598 | 0.6795 | 28.2847 | 14.8086 | 10.3559 | 11.2835 | 14.6062 |
| 0.9571 | 0.5940 | 0.5346 | 0.6246 | 0.6698 | 0.6164 | 37.9375 | 44.1438 | 34.7404 | 30.0178 | 35.5971 |
| 0.7576 | 0.7611 | 0.6095 | 0.5959 | 0.6398 | 0.4928 | 0.4620 | 19.5486 | 21.3437 | 15.5491 | 34.9525 |
| 0.6277 | 0.6921 | 0.5654 | 0.6935 | 0.6802 | 0.6795 | 10.2597 | 9.9251 | 10.4827 | 8.3639 | 8.2523 |
| 0.8896 | 0.6743 | 0.5290 | 0.7551 | 0.7353 | 0.7422 | 24.2019 | 40.5351 | 15.1192 | 17.3449 | 16.5692 |
| Mean: | | | | | | 19.7821 | 33.9239 | 22.9396 | 21.8496 | 22.9909 |
| Maximum: | | | | | | 57.6058 | 151.9352 | 93.1593 | 78.7129 | 68.1818 |
| Minimum: | | | | | | 0.3881 | 2.9107 | 1.8399 | 3.7649 | 3.0670 |

## 6. Conclusions

AOD inversion based on remote sensing technology is being increasingly used for air quality research and is

important for monitoring and predicting air quality at a large scale. The proposed PM2.5/PM10 ratio reflects the air
quality and impact of human activities, which is strongest in winter and summer and weakest in spring and autumn. In
this paper, we used the DDV method to invert the 59 AOD data points in Wuhan in 2017 based on MODIS images. After
the AOD was corrected by the PBLH and RH, the AOD*, which had a greater correlation with PM2.5/PM10, was
obtained, which indicated that the method of correction with the PBLH and RH was effective. After combining gas



pollutants and meteorological data, the optimal subset method was used to find the set of factors that were most suitable
for the prediction of PM2.5/PM10. Since the LSTM model uses the gates as switches, better PM2.5/PM10 prediction
results can be obtained. We hope to obtain a model that can predict air pollution anytime and anywhere by means of
relative factors. Therefore, we set up three prediction patterns: time, space and random patterns. Among the five
intelligent models for comparison, the LSTM model was the most effective, followed by the SVM model, and the
CHAID decision tree model was the least effective. The relatively good results of the LSTM model were reflected not
only in a higher average prediction accuracy but also in the better prediction of maximum and minimum values.
Moreover, the accuracy of the LSTM model was more stable. However, the predictions for the maximum and minimum
values were always below average, which will be the next focus of improvement.
**Code availability** Code content can be accessed through the following website: https://data.mendeley.com/datasets/zk9k53zw3z/1
**Data availability** Experimental data can be accessed through the following website: https://data.mendeley.com/datasets/zk9k53zw3z/2
**Author contribution** All authors worked collectively. Xueling Wu contributed to the conception of the study; Ying Wang contributed
to analysis and manuscript writing; Siyuan He helped perform the analysis with constructive discussions; Zhongfang Wu performed
the data analyses.
**Competing interests** The authors declare that they have no conflict of interest.
**Acknowledgements** This study was jointly supported by the National Natural Science Foundation of China (41871355 and

341 41571438).

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
