# Peer review of "PM2.5/PM10 Ratio Prediction Based on a Long Short-term Memory Neural"

_Geoscientific Model Development, 2019_

## Referee Comment (RC1) · Anonymous Referee #1 · 4 Sep 2019

Reviewers' comments: The manuscript "PM2.5/PM10 Ratio Prediction Based on a Long Short-term Memory Neural Network in Wuhan, China" aims at improving the accuracy and spatio-temporal applicability of PM2.5/PM10 by using long short-term memory (LSTM) neural network method based on the corrected AOD, meteorological data and gaseous pollutant data in Wuhan. The results showed that the LSTM model had significant advantages compared with the results of the four other intelligent models. This is a very interesting study. However, there are some problems need to be revised very carefully. For now, it cannot be accepted. Abstract: The MODIS first appeared in keywords, so Moderate Resolution Imaging Spectroradiometer should provide abbreviations in the abstract. Introduction 1. The introduction needs significant improvement.

Please following the state of the paper, which is nor presented well in this paper. Why do authors want to conduct this study? What is the significance of this research? The introduction only introduced the research status, advantages of methods and models, and research purposes, but does not specify the significance of this study. 2. In the introduction, you described PM10 and PM2.5. Why does the paper only point out aerodynamic particle size of PM2.5, but not PM10? 3. PM10 is primarily produced by natural processes, such as resuspending local soils, sandstorms, and roadside dust, and various industrial processes. This sentence need significant improvement. PM10 is not only derived from natural processes, but also from anthropogenic emissions. In addition, were the roadside dust and various industrial processes generated by natural processes? 4. In the lines 30-31, "are particularly important for environmental policy and public health research". What does this sentence focus on? Anthropogenic combustion products? Please revise carefully. 5. In the lines 36-38, this sentence lacks the reference. Some other sentences lacking references in the introduction. Please check them carefully. Besides, some references are too old in the introduction. 6. "Many statistical models have been used for the ground PM estimation of AOD and other predictors, such as linear regression models, random forest models, neural network models, and generalized additive models." In this sentence, each model should be added with the corresponding reference. 7. Overall, the introduction is too long. Because the introduction does not mean the stack of the references, such as the lines 78-81, and there are similar problems in some sentences in the introduction. In addition, there is a suggestion that the second and third paragraphs should be selectively integrated. Methods: 1. In the line 141, why do gaseous pollutants exist subscripts and particulate matter do not? 2. In the lines 143-162 of the section 3.2, are the sources of the gaseous pollutants described in this section the results generated by authors? If not, please add the references to support your statements. 3. In section 3.3, a total of 5 meteorological stations exist near the Wuhan area. Are there too few stations for interpolation? Please provide parameters to prove that the spherical model of the kriging method used in the paper is reasonable. 4. Sections 3 and section 4 are methods,

and they should be integrated together. Results and discussion Results and discussion section, it reads like just the results and there is little discussion. It is suggested that this paper should be compared with other articles on neural network models to ensure the credibility and stability of the results.

---

## Short Comment (SC1) · 4 Sep 2019

The study proposed a novel framework to predict PM2.5/PM10 ratio based on satellite AOD retrievals. This research idea is interesting and very important, and the authors have made great efforts in developing the technique.

---

## Short Comment (SC2) · 8 Sep 2019

This paper is well organized and structured. Only a few places need to be addressed after the review. Please see the following comments.

**64 Add 2-3 references after this sentence to support this statement. #69 "better" is too general, pick a specific word (e.g., more precise, accurate, efficient, reliable) #73 The same, need to add 2-3 references after this sentence to support this statement. #82 Add one sentence to emphasize the importance/significance of your research, to tell why it's worth studying. Something like "In order to overcome/ improve ...from previous research, this paper utilize ..." #89 How's the prediction result compared to**

other methods? Any effectiveness test to prove that your prediction is more accurate and more reliable? #105 (1) use different sizes of annotation to distinguish different administrative level (e.g., Hubei Province, Wuhan City). (2) why highlight Beijing as it is not your study area? If so, you should add one locational sentence mention that like "Wuhan is located on the south of Beijing, the capital of China....") #113 Add some general description for the data you used in the paper although you introduced the details in the following paragraph (e.g., time period) #118 Add extra sub-blocks for your flowchart by using dash line blocks (data preprocessing step, data analyzing step, data prediction step, validation step...) #134 change the font for "lookup table" #146 add more quantitative description of the data by comparing different periods to show the difference among seasons. For example, PM2.5, summer indicators only accounted for one-third or even one-fourth of the winter indicator values. The same for other indicators, you have to not only showing the data but also mining them. #162 Add Wuhan to the table name. #171 If possible, show why the methods you picked are better than others instead of repeating what is Kriging method. #247 Since you mentioned the LSTM model is good at dealing data with long intervals and long delays, please list a few application fields to show its advantages over RNN, HMM or other models. add more references here, one is not enough to support the statement. #268 Add one paragraph here to generally describe your LSTM model in order to smoothly connecting to the details in the following paragraphs. #294 Need to add a summary to your "Time pattern prediction" analysis. This sub-summary targets to explain/make clear this sub-section result. ps: this part is different from the conclusion, which is the sublimation of the entire paper. #308 the same, need to add a sub-summary to your Time Spatial pattern prediction. see my comments @ line 294 #317 the same, need to add a short summary to your Random pattern prediction. See my comments @ line 294 #327 Avoid using the word "hope", this is the conclusion part, you are done with the analysis, you have to tell the reader about your contribution and possible future work of related study.

Please also note the supplement to this comment:
https://www.geosci-model-dev-discuss.net/gmd-2019-180/gmd-2019-180-SC2-supplement.pdf

———————————————————

---

## Short Comment (SC3) · 23 Sep 2019

This paper proposes a PM2.5/PM10 Ratio Prediction Based on a Long Short-term Memory Neural Network methodïijŇand carries out simulation analysis of measured data with Wuhan as the main research area. This study used 9 main factors to predict PM2.5/PM10 based on time space, and random patterns, and compared the LSTM model with other intelligent models. The results showed that the LSTM model had significant advantages in the study of PM2.5/PM10 prediction, which provides an excellent idea for the research of air pollution monitoring and forecasting in China, and contributes to the application of machine learning in this field.

---

## Short Comment (SC4) · 24 Sep 2019

This paper proposes a PM2.5/PM10 Ratio Prediction Based on a Long Short-term Memory Neural Network methodïijNand carries out simulation analysis of measured data with Wuhan as the main research area. This study used 9 main factors to predict PM2.5/PM10 based on time space, and random patterns, and compared the LSTM model with other intelligent models. The results showed that the LSTM model had significant advantages in the study of PM2.5/PM10 prediction, which provides an excellent idea for the research of air pollution monitoring and forecasting in China, and contributes to the application of machine learning in this field. Besides, I suggest that

authors should focus on one key points and make a deeper disscusion, models or air pollution, bite off more than one can chew only led a long literary piece, but superificiality research.

———————————————————

---

## Short Comment (SC5) · 25 Oct 2019

This is a comment concerning the compliance of this manuscript with GMD standards for code and data availability. The issues raised here must be addressed before a revised manuscript can be published.

This is a slightly unusual manuscript by GMD standards, because the model is a trained neural net using off-the shelf technology provided by Keras. There is absolutely no issue with this scientifically, but from a code and data availability perspective, it is necessary to read our rules in the spirit in which they are intended: which is that all of the data needed to reproduce the manuscript needs to be archived.

[Figure]

The issues are:

1. Preprocessing code is missing. The manuscript details extensive preprocessing which was applied to the data. The code and data presented needs to represent all stages of the work, not just the final neural net. Please include the raw data and the code which preprocesses it to create the training set.

2. Evaluation code is missing. The code used to conduct the evaluation experiments and in particular to produce tables 5, 6, and 7. The interested reader should be able to run a script and have the evaluation occur, resulting in the data in those tables. This is a critical part of the reproducibility of the paper.

3. Missing spreadsheets. The Python script provided loads two Excel sheets of data. These sheets are not in the archive, so users can't actually repeat your work. The spreadsheets need to be included (I presume they contains the same data as the supplied PDF, but that doesn't really help someone wanting to run the code).

4. The work is critically dependent on the LSTM implementation in Keras. It is therefore necessary to specify the exact version of Keras which was used to undertake the work, in case a different version produces different results.

5. Don't use URLs to refer to data on Mendeley. The record on Mendeley even tells you how to cite them. You need to put a reference to the data in your bibliography, including the DOI, and cite that from the code and data section.

---

## Author Comment (AC1) · 22 Nov 2019

We would like to thank all of the positive and constructive comments concerning our revised manuscript. These comments are all valuable and helpful for revising and improving our paper. We have carefully studied the comments. Based on the comments, we have made corrections, and in the revised manuscript which in the supplement, all the corrections were marked in red. Point-by-point responses to the comments:

1. Comment: Abstract: The MODIS first appeared in keywords, so Moderate Resolution Spectroradiometer should provide abbreviations in the abstract.

[Figure]

Response: Thank you for pointing this out. We agree with this valuable comment. The correction is as follows (line 12-14):

First, the aerosol optical depth (AOD) in 2017 in Wuhan was obtained based on Moderate Resolution Imaging Spectroradiometer (MODIS) images, with a 1 km spatial resolution, by using the Dense Dark Vegetation method. Second, the AOD was corrected by calculating the planetary boundary layer height and relative humidity.

2. Comment: Introduction 1 The introduction needs significant improvement. Please following the state of the paper, which is nor presented well in this paper. Why do authors want to conduct this study? What is the significance of this research? The introduction only introduced the research status, advantages of methods and models, and research purposes but does not specify the significance of this study.

Response: Thank you for your thoughtful insights. We agree with this valuable comment. We added a description of the purpose of the study in Section 1. The correction is as follows (line 83-86)): At present, air quality monitoring is still mainly based on monitoring stations, and it is difficult to acquire large-scale and accurate prediction results. In order to reduce the dependence on monitoring stations and achieve the goal of broad, rapid and accurate air quality predictions, this paper aims to use a machine learning algorithm, combined with AOD, gaseous pollutant and meteorological data, to obtain a spatially and temporally reliable prediction model.

3. Comment: In the introduction, you described PM10 and PM2.5. Why does the paper only point out aerodynamic particle size of PM2.5, but not PM10? Response: Thank you for your thoughtful insights. We agree with this valuable comment. The correction is as follows (line 27-30):

Particles with an aerodynamic particle size not exceeding 10 $\mu$m are called PM10. PM10 is primarily produced by industrial production, agricultural production, construction, roadside dust, various industrial processes and natural processes such as the resuspension of local soil and dust storms..

4. Comment: PM10 is primarily produced by natural processes, such as resuspending local soils, sandstorms, and roadside dust, and various industrial processes. This sentence need significant improvement. PM10 is not only derived from natural processes but also from anthropogenic emissions. In addition, were the roadside dust and various industrial processes generated by natural processes?

Response: Thank you for your helpful insights. We agree with this valuable comment. The correction is as follows (line 27-30): Particles with an aerodynamic particle size not exceeding 10 $\mu$m are called PM10. PM10 is primarily produced by industrial production, agricultural production, construction, roadside dust, various industrial processes and natural processes such as the resuspension of local soil and dust storms..

5. Comment: In the lines 30-31, "are particularly important for environmental policy and public health research". What does this sentence focus on? Anthropogenic combustion products? Please revise carefully.

Response: Thank you for your helpful insights. We agree with this valuable comment. The correction is as follows (line 31-33): PM2.5 is mainly produced by anthropogenic combustion for transportation and energy production, and it is particularly important in environmental policy and public health.

6. Comment: In the lines 36-38, this sentence lacks the reference. Some other sentences lacking references in the introduction. Please check them carefully. Besides, some references are too old in the introduction.

Response: Thank you for your thoughtful insights. We updated some of the references. Since some of the research results are derived from classic papers, several older references have been retained. The correction is as follows (line 37-39): In addition, since the scattering extinction contribution of PM2.5 particles accounts for 80% of the extinction of the atmosphere, the concentration of PM2.5 is a key factor in determining the visibility of the atmosphere (Sisler and Malm, 1997).

7. Comment: "Many statistical models have been used for the ground PM estimation of AOD and other predictors, such as linear regression models, random forest models, neural net- work models, and generalized additive models." In this sentence, each model should be added with the corresponding reference.

Response: Thank you for your thoughtful insights. The correction is as follows (line 74-76): Many statistical models have been used for the ground PM estimation of AOD and other predictors, such as linear regression models (Kim et al., 2019), random forest models (Stafoggia et al., 2019), neural network models (Sowden et al., 2018), and generalized additive models (Chen et al., 2018).

8. Comment: Overall, the introduction is too long. Because the introduction does not mean the stack of the references, such as the lines 78-81, and there are similar problems in some sentences in the introduction. In addition, there is a suggestion that the second and third paragraphs should be selectively integrated.

Response: Thank you for your thoughtful insights. We agree with this valuable comment. We removed the extra references in lines 80-81 and merged the second and third paragraphs. The correction is as follows (line 52-54): There are many ways to obtain the AOD from satellite sensors such as the Geostationary Operational Environmental Satellites (GOES) (Prados et al., 2007), the Advanced Very High Resolution Radiometer (AVHRR) (Gao et al., 2016), and the Moderate Resolution Imaging Spectroradiometer (MODIS) (Levy et al, 2013).

9. Comment: Methods: 1. In the line 141, why do gaseous pollutants exist subscripts and particulate matter do not?

Response: Thank you for your helpful insights. We are very sorry for our incorrect notation. We modified the particulate matter subscripts throughout the text.

10. Comment: In the lines 143-162 of the section 3.2, are the sources of the gaseous pollutants described in this section the results generated by authors? If not, please add

the references to support your statements.

Response: Thank you for your helpful insights. The monthly average values of gaseous pollutants come from calculations with the daily data released by the China National Environmental Monitoring Center (http://webinterface.cnemc.cn/cskqzlrbxsb2092932.jhtml). We described the data source and re-stated the changes in the data. The correction is as follows (line 149-151): The calculations in this paper were based on these daily averaged data, which were released by the China National Environmental Monitoring Center (http://webinterface.cnemc.cn/cskqzlrbxsb2092932.jhtml).

11. Comment: In section 3.3, a total of 5 meteorological stations exist near the Wuhan area. Are there too few stations for interpolation? Please provide parameters to prove that the spherical model of the kriging method used in the paper is reasonable. Response: Thank you for your thoughtful insights. Since Wuhan is a provincial capital, the number of meteorological stations around Wuhan is higher than that around other cities, and the distribution of the five stations is relatively scattered. The interpolation results from them are reasonable. We also added a description of the kriging interpolation method. The correction is as follows (line 178-188): We believe that the kriging method is the most appropriate for examining the spatial characteristics of meteorological data.. The kriging method is a multi-step process that includes exploratory statistical analysis of the data, variogram modelling, surface creation, and studying the various surfaces. The kriging method interpolates unknown samples according to the distribution characteristics of a few well-known data points in a finite neighborhood. After taking into account the size, shape, and spatial orientation of the sample points, combining the spatial relationship between the known sample points and the unknown samples, and adding the structural information provided by the variogram, kriging performs a linear unbiased optimal estimation of the unknown samples in the spatial range. After comparing the kriging, natural neighbor, spline, and inverse distance weighted methods, we found that the results acquired by setting 12 interpolation

points and using the spherical model of the kriging method were smoother and more suitable for the study area.

12. Comment: Sections 3 and section 4 are methods, and they should be integrated together. Results and discussion Results and discussion section, it reads like just the results and there is little discussion. It is suggested that this paper should be compared with other articles on neural network models to ensure the credibility and stability of the results.

Response: Thank you for your helpful insights. We are very sorry for the incorrect organization. We changed the title of Section 3 to "Data". We added discussion and analysis at the end of each paragraph in Section 5. The corrections are as follows (line 314-322, 327-330, 333-336, 350-353, 369-373): In air quality research, predictions of higher values are particularly important, because only a successful prediction of poor air quality can be used to promptly remind people to take preventive measures, such as wearing masks. This table was produced in site order, i.e., the first and second data entries are from the same site for the last two days of 2017, and the third and fourth data are from another site. The actual data for PM2.5/PM10 on the first day were generally lower than those on the next day, and the data from 7 of the sites on the last day were larger than 0.8. Only the LSTM model could produce predictions at such extremely high values. In the other models, there was only one result greater than 0.8 for the prediction data, while the LSTM algorithm had three prediction results higher than 0.8. This result indicates that LSTM produced better predictions at higher values than the other machine learning model algorithms.

Since the prediction site had no input data for the whole year and is far away from the other 9 stations, the prediction result was less accurate than the time and random prediction results. However, this prediction method can better reflect the applicability of the model to spatial prediction. In this spatial prediction, the accuracy of the prediction result when the PM2.5/PM10 was lower than 0.2 was the lowest, and the accuracy of the prediction result when the PM2.5/PM10 was larger than 0.8 was better than that
when the PM2.5/PM10 was lower than 0.2. The prediction results in other cases were much better.

The random pattern prediction was based on the completely random selection of time and space aspects and can reflect the effect of air quality prediction under various climatic conditions well. The superiority of the LSTM model prediction in the random prediction pattern was more obvious than in the other patterns, which indicates that under irregular conditions, the LSTM model is more suitable for making predictions.

Since LSTM is a time-recurrent neural network that is suitable for processing and predicting events with relatively long intervals and delays in time series, the time pattern prediction results for the three prediction models are the most accurate, and the spatial pattern prediction results without any time data are the least accurate. However, the predictions for the maximum and minimum values were always below average, especially the prediction of the maximum value. The next focuses for improvement will be the optimization of the algorithm and the improvement of the prediction accuracy.

Please also note the supplement to this comment:
https://www.geosci-model-dev-discuss.net/gmd-2019-180/gmd-2019-180-AC1-supplement.pdf

——————————————————

[Figure]

**Supplement:**

[revised manuscript text omitted]

74 Many statistical models have been used for the ground PM estimation of AOD and other predictors, such as linear 75 regression models (Kim et al., 2019), random forest models (Stafoggia et al., 2019), neural network models (Sowden et 76 al., 2018), and generalized additive models (Chen et al., 2018). However, with the introduction of new intelligent models, 77 the traditional regression model reflects the inability to balance time, space and random precision. One way to overcome 78 these limitations is the long short-term memory (LSTM) model. The LSTM network is ideal for learning from experience 79 so that time series can be classified, processed, and predicted with very long unknown time lags between important 80 events. In the study of PM2.5 monitoring and prediction in smart cities, Chiou-Jye et al. proposed that the prediction 81 accuracy of the convolutional neural network (CNN)-LSTM model is the highest compared to the prediction accuracies 82 of several other classic machine learning methods (Chiou-Jye and Ping-Huan, 2018).

At present, air quality monitoring is still mainly based on monitoring stations, and it is difficult to acquire large-scale and accurate prediction results. In order to reduce the dependence on monitoring stations and achieve the goal of broad, rapid and accurate air quality predictions, this paper aims to use a machine learning algorithm, combined with AOD, gaseous pollutant and meteorological data, to obtain a spatially and temporally reliable prediction model. This paper used a total of 59 AOD results for all of 2017 by the Dense Dark Vegetation (DDV) method using MODIS level-2 data of Wuhan with a spatial resolution of 1 km. Since there were only 10 air quality stations in Wuhan, to ensure accuracy, the AOD values were extracted at the air quality station site, and the integration of the AOD, air pollutants, and 90 meteorological data was also based on the station site. AOD\* was obtained by correcting AOD using the PBLH and RH. 91 Then, the  $R^2$ -based optimal subset selection method was used to select the most relevant factor for  $PM_{2.5}/PM_{10}$  from the 92 meteorological factors and air pollutants. Finally, the space and time scales and random  $PM_{2.5}/PM_{10}$  predictions were 93 determined and performed, respectively, via the LSTM model, and the prediction results of the LSTM model and other 94 classic models were compared and analysed. The results showed that the average error of the LSTM model prediction 95 results is very low, both spatially and temporally, and the stability of the prediction model is significantly better than that 96 of other models.

**97 **2.** Study area**

98 Wuhan is the provincial capital of Hubei Province. The administrative extent is between 113.683°E-115.083°E and 29.967°N-31.367°N, and the total area is 8494.41 km2 (Zhou and Chen, 2018). The largest distance is between the 99 100 eastern and western parts of Wuhan and is 134 km, and the maximum distance from north to south is 155 km. Wuhan is 101 the city with the largest population, is the largest provincial capital city, has the most complicated road traffic and has the 102 most developed economy in the central part of the country (Jiao et al., 2017). The Yangtze River flows through Wuhan, 103 and there are hundreds of lakes in Wuhan. The terrain of Wuhan is mainly plains, with low levels in the middle of the 104 region and low mountains, hills and ridges to the south and north. The climate type is a humid, north subtropical 105 monsoon climate with high temperatures in summer, low temperatures in winter, and an annual average temperature of 106 15.9 °C. Sunshine hours and total radiation are also at high levels, and the annual average precipitation is approximately 107 1300 mm. June and August receive the most precipitation in Wuhan, and summer precipitation accounts for 108 approximately 40% of the annual rainfall. In recent years, the air quality in Wuhan has been improved. In 2017, the 109 number of days in which the annual air quality level was acceptable was 255 days, and the acceptability rate was 69.9%. 110 At the same time, the number of days with light pollution, moderate pollution, heavy pollution, and severe pollution was 111 86 days, 17 days, 6 days, and 1 day, respectively.

113 Fig. 1 Location of the study area in China (A: map of China, B: map of Wuhan).

**114 **3. Data**

115 The data that our environmental monitoring station can monitor are only real-time data. If we want to predict the 116 state of the air afterwards, we can use other relevant factors for reference. The AOD is an important parameter in the 117 study of atmospheric aerosols, which have a great relationship with PM. Gaseous pollutants are also a key factor in air 118 quality. In addition, changes in meteorological conditions have an impact on PM. Therefore, we used the air quality data 119 from the ground monitoring station as the inspection standard and extracted the values of these correlation factors with 120 the data from the monitoring site for verification. After retrieving the AOD with the MODIS images five times a month, 121 on average, in 2017, the AOD values at the monitoring site were extracted, and the values of the meteorological data 122 were also interpolated at the same point. Then, the AOD was corrected to obtain the AOD\*, and gaseous pollutant data at 123 the monitoring site were added. The best set that predicted air quality was selected, and machine learning techniques 124 were used to obtain models that can make space and time series predictions (Fig. 2).

---

## Referee Comment (RC2) · Anonymous Referee #2 · 11 Dec 2019

This paper is addressing an important research question. It is generally well written. However, some important points need clarifying before publication.

In particular, the authors should clarify the way they split their data. They state on line 271, "80% of the data were used as the training sample for modelling, and 20% of the data were used as the verification sample." It is important to specify how the hyper-parameters of their model were chosen. If they were chosen by optimising the performance against the verification dataset then it is possible that the algorithm has over-fit the hyper-parameters. Lines 272-274 seem to imply there was at least some hyper-parameter tuning performed. Ideally, the data should be split three ways, into

a training, verification, and test, so that the hyper-parameters are tuned against the verification data, and the algorithm scored against the test data. The authors should reassure the reader that they have taken measures to ensure they have not overfit the hyper-parameters - for instance, perhaps they further split their training set.

In addition, they state on line 279 that for Section 5.1 that there were 570 samples in the training data, and what I infer is 20 samples in the verification data (two days multiplied by ten sites, as per the training data). This appears not to be an 80%/20% split. In any case, the verification data are from one period in the season (end of December) - the algorithm may simply be good at predicting air quality in December but not the rest of the year. A more convincing approach would be to test against multiple cases from throughout the year. This is similar for the spatial prediction, which appears to only be tested at one site.

As such, the authors must reassure us that their approach to validation guards against overfitting in order for this to be suitable for publication.

A series of specific comments and questions now follows:

* Could you discuss why you have focussed on the PM2.5/PM10 ratio as opposed to considering them separately?

* line 75: "random precision", and Section 5.3 "random pattern prediction". Please could you clarify what this is - it was unclear to me. Are you randomly selecting a subset of points in space and then predicting them with the remaining, contemporaneous points? If so, how is this significantly different from the spatial prediction in Section 5.2? Please better explain this task near the beginning of the manuscript.

* line 112,113: I found this sentence confusing. Are "monitoring station" and "monitoring site" different things? I'm unclear what the definition is on an "inspection standard"? Does this mean the "truth" data you are using for the verification. I'm unclear what "correlation factors" means.

[Figure]

* line 132: You verify your data processing against NASA data. Is NASA has a product, why not just use that?

* line 175: "higher trend" and "lower trend". The word "trend" changes the meaning. I presume this should read "average temperature is higher in summer and lower in winter", as expected. Otherwise, I don't understand what it means.

* line 185: Is this standard published. If so, please cite. In my opinion, this can be a technical paper as opposed to a peer review paper (but the editor may feel differently).

* line 230: Could you clarify the optical subset approach? Was the R^2 score performed on the output of the LSTM as compared with the observations. If so, it may be better to put this section after the description of the LSTM and make that clear.

* Table 4: I presume this table shows only the top 10 scoring selections? Presumably you scored all combinations of predictors. Please explain.

* Line 258: Normally the first gate is expressed at deciding what to forget, rather than what to remember (I appreciate they are equivalent). Figure 7 shows a "Forget gate" so it would be helpful to standardise the terminology.

* "and the third switch controls whether c is the current output of the LSTM model" I'm not sure about this - correct me if I'm wrong, but isn't $c_t$ combined with $h_{t-1}$, and $x_t$ to create $h_t$, which is the output?

* Section 4: I see the link to your code, that you used Keras and their LSTM implementation, which is great. If possible, could you cite Keras directly in the manuscript, and state that you used their implementation of LSTMs.

* line 273: "with the first three layers being the LSTM layer and the last layer being the dense layer". I presume this means that you have three LSTM units follows by a dense layer. However, each LSTM unit can be thought of of comprising multiple layers, so this terminology is confusing. In addition, could you explain to the reader the purpose of the final dense layer.

* line 282: I don't understand the difference between a "multilayer perceptron" and a "artificial neural network". In addition, I would have thought the LSTM, multilayer perceptron and artificial neural network, all rely on back propagation, so I don't understand the terminology "back propagation artificial neural network". Please clarify.

* Section 5.3: as mentioned above, I don't understand what is being done in this section.

Typographic and language errors:

* line 24: "Aerosols are a general term" -> "Aerosol is a general term"

*Please define all your acronyms on first use, for instance RH, DVV etc

* line 74: "intelligent models", please clarify what this means

* line 89: "classic" -> "classical"

* line 108: -> "Our environmental monitoring station only monitors data in real-time". I'm also not sure what this sentence is meant to mean. Are you saying that the instrument doesn't provide information about the future? Please clarify.

* line 110: "which have", is ambiguous - is this referring to the AOD or the atmospheric aerosols? It seems redundant to say that PM is linked to atmospheric aerosols. Please clarify this sentence.

* line 140: "The shortest distance between points exceeds 3 km, and the average distance exceeds 10 km." I don't know what the word "exceeds" means in this context. Can we replace it with "is" instead?
* * *

---

## Author Comment (AC2) · 16 Dec 2019

Manuscript number: gmd-2019-180 Xueling Wu *, Ying Wang, Siyuan He, Zhongfang Wu: "PM2.5 / PM10 Ratio Prediction Based on a Long Short-term Memory Neural Network in Wuhan, China". Dear editor and reviewer, We would like to thank you for the positive and constructive comments concerning our revised manuscript (ID: gmd-2019-180). These comments are all valuable and very helpful for revising and improving our paper. We have studied the comments carefully. Based on the comments, we have made corrections which we hope will meet with your approval. In the revised manuscript, all the corrections are marked in red. The responses to the reviewers'

comments are as follows (in blue font):

Responses to the Referee comment 2:

Point-by-point responses to the comments: 1. Comment: In particular, the authors should clarify the way they split their data. They state on line 271, "80% of the data were used as the training sample for modelling, and 20% of the data were used as the verification sample." It is important to specify how the hyper-parameters of their model were chosen. If they were chosen by optimising the performance against the verification dataset then it is possible that the algorithm has over-fit the hyper-parameters. Lines 272-274 seem to imply there was at least some hyper-parameter tuning performed. Ideally, the data should be split three ways, into a training, verification, and test, so that the hyper-parameters are tuned against the verification data, and the algorithm scored against the test data. The authors should reassure the reader that they have taken measures to ensure they have not overfit the hyper-parameters - for instance, perhaps they further split their training set. In addition, they state on line 279 that for Section 5.1 that there were 570 samples in the training data, and what I infer is 20 samples in the verification data (two days multiplied by ten sites, as per the training data). This appears not to be an 80%/20% split. In any case, the verification data are from one period in the season (end of December)-the algorithm may simply be good at predicting air quality in December but not the rest of the year. A more convincing approach would be to test against multiple cases from throughout the year. This is similar for the spatial prediction, which appears to only be tested at one site.

As such, the authors must reassure us that their approach to validation guards against overfitting in order for this to be suitable for publication. Response: Thank you for pointing this out. After considering your suggestions, we readjusted our model and code. Except for the data used for prediction, we divided the data set involved in the model construction into three parts: 40% of the data were used as the training samples for modeling, 30% of the data were used as the test samples, and the remaining 30% of the data was used as verification data. We tried to add a regularization term, but the

effect did not improve. After adjusting the number of neurons, the number of epochs, and the batch size, the loss function we obtained has converged without overfitting. Moreover, the revised model obtained higher prediction accuracy than the original one. We have generated the following learning curves for three prediction patterns, but in order to avoid the manuscript being too verbose, we do not intend to add the learning curve to the manuscript. The three curves in this figure are the losses of training samples, test samples and verification samples (train, test, and verifi) that increase with the number of epochs. We understand that our spatial and time prediction patterns do not completely cover the whole year and all regions, so we have added a random prediction pattern which randomly selects data from the whole year and the entire region for prediction to reduce fortuity of the other two prediction patterns. The correction is as follows (line 304-313):

Figure 1 Learning curves To determine the appropriate number of layers for the LSTM method, except for the data used for prediction, we divided the data set involved in the model construction into three parts: 40% of the data were used as the training samples for modeling, 30% of the data were used as the test samples, and the remaining 30% of the data was used as verification data. We tried to use various LSTM architecture layers for the comparison. After obtaining the results of various LSTM architecture layers, we found that the results obtained using the LSTM architecture with four layers were the best, with the first three layers and the dense layer as the last layer. The role of the dense layer is to complete the final output of unique values. Because the LSTM uses the activation function as the gate, the outputs of the gates must be between 0 and 1, and the output ranges of both types of activation functions must be satisfied. We determined that the activation function for setting the forget gate and the input gate was defined as a sigmoid function. After adjusting the number of neurons, the number of epochs, and the batch size, the loss function we obtained has converged without overfitting.

2. Comment: Could you discuss why you have focused on the PM2.5/PM10 ratio as

opposed to considering them separately? Response: Thank you for pointing this out. The explanation of using the PM2.5-PM10 scale is as follows (line 43-47): Since fine and coarse particles come from different sources, the PM2.5-PM10 scale model has different physicochemical properties, which can not only distinguish the type of aerosol in the PM but also provide the mixing ratio of dust and artificial aerosols (Sugimoto et al., 2015). The PM2.5-PM10 scale is the main indicator for macro analysis of the source of particulate pollution in a region, which is more practical than considering PM2.5 and PM10 separately.

3. Comment: line 75: "random precision", and Section 5.3 "random pattern prediction". Please could you clarify what this is - it was unclear to me. Are you randomly selecting a subset of points in space and then predicting them with the remaining, contemporaneous points? If so, how is this significantly different from the spatial prediction in Section 5.2? Please better explain this task near the beginning of the manuscript. Response: Thank you for pointing this out. We have added the explanation of these three prediction patterns to the Section 1-Introduction (line 80-84): The time precision mentioned in this article refers to the accuracy of inputting time-series data to predict the subsequent period results; the spatial precision refers to the accuracy of inputting all-time data of spatial points to predict the result of another spatial point; the random accuracy refers to the accuracy of inputting data of any time and space to predict the random selection data.

4. Comment: line 112,113: I found this sentence confusing. Are "monitoring station" and "monitoring site" different things? I'm unclear what the definition is on an "inspection standard"? Does this mean the "truth" data you are using for the verification. I'm unclear what "correlation factors" means. Response: Thank you for your thoughtful insights. "Monitoring station" and "monitoring site" have the same meaning, and we replaced "monitoring site" with "monitoring station". "Inspection standard" means truly data and "Correlation factors" refer to PM2.5, PM10, and gaseous pollutant data detected by the stations. We have redefined these meanings in the article (line 126-127):

[Figure]

Therefore, we used the truly air quality data from the ground monitoring stations as the inspection standard for verification and extracted the values of PM2.5, PM10, and gaseous pollutant with the data from the monitoring stations.

5. Comment: line 132: You verify your data processing against NASA data. Is NASA has a product, why not just use that? Response: Thank you for your thoughtful insights. The commonly used aerosol automatic observation network AERONET jointly established by NASA and CNRS is of good quality and easy to obtain, but the number of stations is limited, and there is no station coverage in the study area. The remote sensing data we collected are better in time and space continuity, and the AOD retrieval algorithm is also applicable to the study area.

6. Comment: line 175: "higher trend" and "lower trend". The word "trend" changes the meaning. I presume this should read "average temperature is higher in summer and lower in winter", as expected. Otherwise, I don't understand what it means. Response: Thank you for your thoughtful insights. We have modified the expression of this sentence (line 197-198): The average surface temperature and average temperature were higher in summer and lower in winter.

7. Comment: line 185: Is this standard published. If so, please cite. In my opinion, this can be a technical paper as opposed to a peer review paper (but the editor may feel differently). Response: Thank you for your thoughtful insights. This standard has been published, so we added a citation as follows (line 206-208): The national standard method is performed according to the method specified in the Chinese national standard GB/T13201-91 (http://www.mee.gov.cn/gzfw_13107/kjbz/qthjbhbz/qt/201605/t20160522_342349.shtml).

8. Comment: line 230: Could you clarify the optical subset approach? Was the RËE̞2 score performed on the output of the LSTM as compared with the observations. If so, it may be better to put this section after the description of the LSTM and make that clear. Response: Thank you for your thoughtful insights. We clarified the optical

subset approach. In addition, we did not perform RЁĘ2 scoring on the output of the LSTM, because the relative error rate can also reflect the accuracy intuitively. Since the maximum relative error and minimum relative error needs to be analyzed at the same time, it is more neatly to display the three relative error rates in a table. The explanation of optimal subset method is as follows (line 249-252): The process of the optimal subset method is that in a set containing multiple independent variables, freely selecting and combining from each independent variable, combining all independent variables and dependent variables to establish all possible equations, and then the best independent variable combination model is selected from all the fitted regression equations.

9. Comment: Table 4: I presume this table shows only the top 10 scoring selections? Presumably you scored all combinations of predictors. Please explain. Response: Thank you for your thoughtful insights. I added an explanation for table 4 as follows (line 257-258): This table shows the top 10 scores for R2 scores and the corresponding factor combinations.

10. Comment: Line 258: Normally the first gate is expressed at deciding what to forget, rather than what to remember (I appreciate they are equivalent). Figure 7 shows a "Forget gate" so it would be helpful to standardise the terminology. Response: Thank you for your thoughtful insights. The correction is as follows (line 286-288): The input gate determines how much of the input xt of the network is saved to the cell state ct at the current moment, the forget gate determines how much the cell state ct-1 at the previous moment is retained to the current moment ct, and the output gate controls how much the cell state ct is output to the current output value ht of the LSTM.

11. Comment: "and the third switch controls whether c is the current output of the LSTM model" I'm not sure about this - correct me if I'm wrong, but isn't c_t combined with h_t-1, and x_t to create h_t, which is the output? Response: Thank you for your thoughtful insights. It is true that ht is created by xt with the combination of ct and ht-1. The correction is as follows (line 282-288): Fig. 6 emphasizes the calculation process

of the cell state c, and the overall process of the LSTM model is shown in Fig. 7. The key point of the LSTM model is how to control the state c. The idea of the LSTM model is to use three control switches to control it. The switches implemented in the algorithm are known as "gates", which are fully connected layers whose input is a vector, and the output is a real vector between 0 and 1 (Srivastava and Lessmann, 2018). The input gate determines how much of the input xt of the network is saved to the cell state ct at the current moment, the forget gate determines how much the cell state ct-1 at the previous moment is retained to the current moment ct, and the output gate controls how much the cell state ct is output to the current output value ht of the LSTM.

12. Comment: Section 4: I see the link to your code, that you used Keras and their LSTM implementation, which is great. If possible, could you cite Keras directly in the manuscript, and state that you used their implementation of LSTMs. Response: Thank you for your thoughtful insights. I cited Keras in the manuscript as follows (line 301-302): The implementation of the LSTM models is based on Keras which is a high-level neural network Application Programming Interface written in Python.

13. Comment: line 273: "with the first three layers being the LSTM layer and the last layer being the dense layer". I presume this means that you have three LSTM units follows by a dense layer. However, each LSTM unit can be thought of comprising multiple layers, so this terminology is confusing. In addition, could you explain to the reader the purpose of the final dense layer. Response: Thank you for your thoughtful insights. This sentence does means that we have three LSTM units follows by a dense layer. To avoid misunderstandings, we adjusted the description and explained the purpose of the dense layer as follows (line 307-310): We tried to use various LSTM architecture layers for the comparison. After obtaining the results of various LSTM architecture layers, we found that the results obtained using the LSTM architecture with four layers were the best, with the first three layers and the dense layer as the last layer. The role of the dense layer is to complete the final output of unique values.

14. Comment: line 282: I don't understand the difference between a "multilayer per-

ceptron" and an "artificial neural network". In addition, I would have thought the LSTM, multilayer perceptron and artificial neural network, all rely on back propagation, so I don't understand the terminology "back propagation artificial neural network". Please clarify. Response: Thank you for your thoughtful insights. The reason why we discriminate between MLP and BP is that MLP is a back propagation neural network model with a three-layer architecture after adjusted in Python by us, while the BP neural network acquired by the clementine software with non-adjustable parameters. After careful consideration, we decided to delete the comparison with MLP to avoid ambiguity.

15. Comment: Section 5.3: as mentioned above, I don't understand what is being done in this section. Response: Thank you for your thoughtful insights. The output of the random prediction pattern in Section 5.3 is randomly selected data in any time and space. This pattern is different from the time and space pattern. The applicability of LSTM reflected by the random prediction pattern is more obvious than the other two patterns. The explanation for section 5.3 is in the manuscript as follows (line 358-362): The random pattern prediction was based on the completely random selection of time and space aspects and can reflect the effect of air quality prediction under various climatic conditions well. The superiority of the LSTM model prediction in the random prediction pattern was more obvious than in the other patterns, which indicates that under irregular conditions, the LSTM model is more suitable for making predictions.

16. Comment: line 24: "Aerosols are a general term" -> "Aerosol is a general term". Response: Thank you for your thoughtful insights. We apologize for our mistakes. The correction is as follows (line 25): Aerosol is a general term for solid and gas particles suspended in air.

17. Comment: Please define all your acronyms on first use, for instance RH, DVV etc. Response: Thank you for your thoughtful insights. The correction is as follows (line 12-15): First, the aerosol optical depth (AOD) in 2017 in Wuhan was obtained based on Moderate Resolution Imaging Spectroradiometer (MODIS) images, with a 1 km spatial resolution, by using the Dense Dark Vegetation (DDV) method. Second,

the AOD was corrected by calculating the planetary boundary layer height (PBLH) and relative humidity (RH).

18. Comment: line 74: "intelligent models", please clarify what this means. Response: Thank you for your thoughtful insights. We replaced "intelligent models" with "machine learning models". The correction is as follows (line 79-80): However, with the introduction of new machine learning models, the traditional regression model reflects the inability to balance time, space and random precision.

19.Comment: line 89: "classic" -> "classical". Response: Thank you for your thoughtful insights. The correction is as follows (line 99-101): Finally, the space and time scales and random PM2.5/PM10 predictions were determined and performed, respectively, via the LSTM model, and the prediction results of the LSTM model and other classical models were compared and analyzed.

20. Comment: line 108: -> "Our environmental monitoring station only monitors data in real-time". I'm also not sure what this sentence is meant to mean. Are you saying that the instrument doesn't provide information about the future? Please clarify. Response: Thank you for your thoughtful insights. The supplement is as follows (line 122-123): The data that our environmental monitoring station can monitor is only real-time data with no predicting subsequent data in advance.

21. Comment: line 110: "which have", is ambiguous - is this referring to the AOD or the atmospheric aerosols? It seems redundant to say that PM is linked to atmospheric aerosols. Please clarify this sentence. Response: Thank you for your thoughtful insights. "Which have" refers to AOD. The correction is as follows (line 123-124): The AOD which has a great relationship with PM is an important parameter in the study of atmospheric aerosols.

22. Comment: line 140: "The shortest distance between points exceeds 3 km, and the average distance exceeds 10 km." I don't know what the word "exceeds" means in this context. Can we replace it with "is" instead? Response: Thank you for your thoughtful

insights. The correction is as follows (line 155): The shortest distance between points is more than 3 km, and the average distance is about 10 km.

A special thanks to you for your insightful and valuable comments.

We greatly appreciate the editors and reviewers for their helpful work and hope that the corrections will meet your approval. Once again, thank you very much for your valuable and helpful comments and suggestions. With best wishes Yours sincerely, Xueling Wu Institute of Geophysics and Geomatics, China University of Geosciences No. 388 Lumo Road, Wuhan 430074, P. R. China Email: snowforesting@163.com

The corrected tables are as follows: Table 5 The results and relative error rates of the time pattern predictions. Measured value Predicted value Relative error rate (%) LSTM BP SVM CHAID LSTM BP SVM CHAID 0.8212 0.6335 0.7786 0.6698 0.4853 22.8604 5.1875 18.4364 40.9036 0.7436 0.5610 0.6961 0.7841 0.4853 24.5491 6.3878 5.4465 34.7364 0.6629 0.7346 0.7074 0.8353 0.6753 10.8069 6.7129 26.0069 1.8706 0.6950 0.7949 0.6850 0.5628 0.6753 14.3746 1.4388 19.0216 2.8345 0.7816 0.7347 0.6871 0.8092 0.5145 5.9982 12.0906 3.5312 34.1735 0.6311 0.7605 0.5864 0.7032 0.6487 20.5089 7.0829 11.4245 2.7888 0.7959 0.7347 0.6870 0.8568 0.6973 7.6931 13.6826 7.6517 12.3885 0.8743 0.8067 0.6474 0.7451 0.6973 7.7307 25.9522 14.7775 20.2448 0.7204 0.6553 0.8106 0.7446 0.8206 9.0291 12.5208 3.3592 13.9089 0.9854 0.7128 0.7154 0.6760 0.8206 27.6610 27.4000 31.3984 16.7242 0.7079 0.7249 0.8321 0.6089 0.7959 2.4048 17.5449 13.9850 12.4311 0.9455 0.7790 0.7064 0.7285 0.7959 17.6108 25.2882 22.9508 15.8223 0.7200 0.4924 0.6692 0.8172 0.6931 31.6131 7.0556 13.5000 3.7361 0.8600 0.6521 0.6192 0.6907 0.6931 24.1694 28.0000 19.6860 19.4070 0.6571 0.6432 0.7011 0.8522 0.5812 2.1242 6.6961 29.6911 11.5508 0.9189 0.7175 0.6195 0.7146 0.5812 21.9150 32.5824 22.2331 36.7505 0.7640 0.7673 0.6549 0.5406 0.7870 0.4291 14.2801 29.2408 3.0105 0.9273 0.7896 0.6354 0.7155 0.7870 14.8513 31.4785 22.8405 15.1299 0.6277 0.4614 0.7308 0.5392 0.6951 26.4993 16.4250 14.0991 10.7376 0.8896 0.6904 0.6685 0.6694 0.7534 22.3909 24.8539 24.7527 15.3103 Mean: 15.7613

16.1330 17.7017 16.2230 Maximum: 31.6111 32.5824 31.3984 40.9036 Minimum: 0.4319 1.4388 3.3592 1.8706

Table 6 The results and relative error rates of the spatial pattern prediction. Models LSTM BP SVM CHAID Mean: 27.9231 34.1333 34.0207 33.7718 Maximum: 178.0639 222.9295 204.7317 230.1367 Minimum: 0.0764 0.1124 0.9026 0.2396

Table 7 The results and relative error rates of the random pattern prediction. Measured value Predicted value Relative error rate (%) LSTM BP SVM CHAID LSTM BP SVM CHAID 0.5870 0.6031 0.5762 0.6091 0.4928 2.7428 1.8399 3.7649 16.0477 0.6213 0.6581 0.6561 0.6826 0.6795 5.9231 5.6012 9.8664 9.3675 0.9843 0.4662 0.6247 0.6185 0.7422 52.6364 36.5336 37.1635 24.5962 0.8000 0.4198 0.4772 0.5231 0.4928 47.5250 40.3500 34.6125 38.4000 0.4638 0.4654 0.4773 0.5136 0.4928 0.3450 2.9107 10.7374 6.2527 0.7010 0.5762 0.6811 0.6675 0.6795 17.8031 2.8388 4.7789 3.0670 0.2222 0.2470 0.4292 0.3971 0.3737 11.1611 93.1593 78.7129 68.1818 0.5929 0.6418 0.6543 0.6598 0.6795 8.2476 10.3559 11.2835 14.6062 0.9571 0.5875 0.6246 0.6698 0.6164 38.6167 34.7404 30.0178 35.5971 0.7576 0.7095 0.5959 0.6398 0.4928 6.3490 21.3437 15.5491 34.9525 0.6277 0.6368 0.6935 0.6802 0.6795 1.4497 10.4827 8.3639 8.2523 0.8896 0.6508 0.7551 0.7353 0.7422 26.8435 15.1192 17.3449 16.5692 Mean: 18.3036 22.9396 21.8496 22.9909 Maximum: 52.6364 93.1593 78.7129 68.1818 Minimum: 0.3450 1.8399 3.7649 3.0670

Please also note the supplement to this comment: https://www.geosci-model-dev-discuss.net/gmd-2019-180/gmd-2019-180-AC2-supplement.pdf
* * *